# Multimodality Imaging in Patients with Hypertrophic Cardiomyopathy and Atrial Fibrillation

**DOI:** 10.3390/diagnostics13193049

**Published:** 2023-09-25

**Authors:** Hyemoon Chung, Eui-Young Choi

**Affiliations:** 1Division of Cardiology, Department of Internal Medicine, Kyung Hee University Hospital, Kyung Hee University, Seoul 02447, Republic of Korea; bluesunny52@gmail.com; 2Division of Cardiology, Heart Center, Gangnam Severance Hospital, Yonsei University College of Medicine, Seoul 06273, Republic of Korea

**Keywords:** hypertrophic cardiomyopathy, multimodality imaging, atrial fibrillation

## Abstract

Ventricular hypertrophy is associated with diastolic dysfunction, resulting in increased left atrial (LA) pressure, enlargement, fibrosis, and decreased LA function. Hypertrophic cardiomyopathy (HCM) is characterized by myocyte disarray, myocardial fibrosis, and hypertrophy. Notably, a thickened and noncompliant LV results in the impairment of diastolic function. These conditions promote LA remodeling and enlargement, which contribute to developing and maintaining atrial fibrillation (AF). AF is an atrial arrhythmia that occurs frequently in HCM, and evaluating the morphology and physiology of the atrium and ventricle is important for treatment and prognosis determination in HCM patients with AF. In addition, it provides a clue that can predict the possibility of new AF, even in patients not previously diagnosed with AF. Cardiac magnetic resonance (CMR), which can overcome the limitations of transthoracic echocardiography (TTE), has been widely used traditionally and even enables tissue characterization; moreover, it has emerged as an essential imaging modality for patients with HCM. Here, we review the role of multimodal imaging in patients with HCM and AF.

## 1. Introduction

Hypertrophic cardiomyopathy (HCM) is an inherited cardiomyopathy caused by mutations in genes encoding sarcomere proteins, affecting approximately 1 in 500 patients in the general population [1]. Myocyte disarray, myofibril disruption, and increased interstitial space with fibrosis result in LV diastolic dysfunction and provide a substrate for supraventricular and ventricular arrhythmias. Moreover, LV diastolic dysfunction and corresponding atrial remodeling contribute to atrial fibrillation (AF) [2], making it a frequent supraventricular arrhythmia in patients with HCM. Notably, AF is an important issue in HCM because it worsens the clinical prognosis by increasing the risk of future stroke and heart failure [3]. In HCM, the ventricular fibrotic burden can contribute to AF generation by greatly influencing LV diastolic and atrial functions. In addition, a recent study showed that inborn atrial myopathy also contributes to the development of AF; therefore, evaluation of the ventricles and atrium is important. 

This review aimed to focus on what should be evaluated using multimodal imaging, including echocardiography and cardiac magnetic resonance (CMR) in terms of AF prediction and disease prognostication in the HCM population on the basis of LA and LV assessment. 

## 2. AF in HCM

AF is the most common arrhythmia in patients with HCM, occurring in approximately 20–30% of previous HCM cohorts [4,5,6]. A study reported that HCM patients with AF have a higher risk of developing a left atrial (LA) thrombus than non-HCM patients do, which further increases the risk of stroke [7]. 

LA remodeling is a major echocardiographic predictor for AF [8,9]. In HCM, impaired ventricular diastolic function due to reduced LV compliance results in elevated LV end-diastolic pressure and LA remodeling. Furthermore, long-standing pressure and volume overload cause structural remodeling, leading to LA dysfunction. However, irrespective of the loading conditions, LA dysfunction can also be caused by a myopathic process in HCM, which is thought to occur because HCM is a genetic disease [10,11]. In addition, evaluation of LA function in patients with AF is important because LA function also increases stroke risk in patients with AF [12,13]. Therefore, evaluating the degree of LA remodeling and LA function is important in cardiac imaging when AF is accompanied in HCM patients [8,9,12]. In patients with LA enlargement or dysfunction, even if AF has never been documented, more intensive monitoring is needed to determine the presence of hidden AF. 

LV remodeling and impaired LV function are associated with AF in patients with HCM. Although primary impairments of sarcomeres caused by sarcomere gene variants are associated with the progression of LV remodeling, increased oxidative stress can also contribute to secondary alterations of sarcomeres. Oxidative stress strengthens sarcomere mutation, and mutant sarcomere proteins become a source of reactive oxygen species, forming a cycle that worsens each other [14]. Oxidative stress results in inflammatory changes and contributes to altered ion channel activity via various pathways, and AF may activate different oxidative pathways, resulting in increasing oxidative stress [15]. Oxidative stress can contribute to both the progress of HCM and the occurrence and maintenance of AF by affecting cardiac remodeling. Diastolic dysfunction increases atrial afterload, stretching, and wall stress, shortening the atrial refractory period and promoting AF occurrence and maintenance by promoting atrial fibrosis and remodeling. One interesting observation is that liver cirrhosis has a protective effect against AF, despite the significant metabolic abnormalities, inflammatory syndrome, and enlarged LA. The low prevalence of AF may be the result of the accumulation of anti-arrhythmic or anti-inflammatory substances that are normally metabolized by an intact functioning liver; this would explain the development of AF after liver transplantation. The administration of spironolactone and beta-blockers is also able to reduce atrial excitability [16]. In liver transplantation patients with preoperative diagnosis of HCM, preoperative LA size (measured by echocardiography) and sudden cardiac death risk were not significantly associated with 1-year survival [17].

In addition, left ventricular outflow tract (LVOT) obstruction, systolic anterior motion (SAM) of the mitral leaflet, and combined MR contribute to LA remodeling and the development of AF. LA enlargement is not only LA remodeling itself but also a surrogate that reflects the pathophysiology of LVs, such as increased wall thickness, diastolic dysfunction, LVOT obstruction, and MR. These conditions can be confirmed mainly by echocardiography. 

## 3. Multimodality Imaging in HCM and AF

The atrium and ventricle should be carefully evaluated using imaging modalities in patients with HCM and AF. Accurate information on LA size and function is crucial for clinicians regarding the prognosis and treatment of AF [18,19]. Furthermore, this information can be used to predict future AF events in patients with HCM whose AF has not yet been diagnosed [20].

### 3.1. Transthoracic Echocardiography (TTE)

TTE is the first-choice modality for evaluating cardiac anatomy and function. Cardiac morphology, as well as cardiac chamber size, function, LVOT obstruction, SAM, and MR that cause or maintain AF in HCM, can also be evaluated using TTE. 

#### 3.1.1. LA Assessment

The accurate determination of LA size is important for patient risk stratification. LA size is a key predictor of future AF development and clinical outcomes, including stroke, in patients with AF [20,21,22]. The anteroposterior (AP) dimension of the LA has been widely used as a traditional parameter to assess LA size. In the ESC guidelines, the LA AP dimension is included in the calculation formula for the sudden cardiac death risk stratification of patients with HCM [23]. However, it is now known that a biplane LA volume by either the disk summation or the area–length method reflects the LA size more accurately than the AP dimension because of LA asymmetry [24]. Previously, Olivotto et al. reported several predictors of AF; the most powerful predictor was the LA dimension, in addition to age and functional class [25]. Subsequently, Tani et al. reported that the LA volume index (LAVI) had better predictive power for AF occurrence than did the LA dimension [26]. In their study, maximum LA volume was the most sensitive and specific predictor of paroxysmal AF [26]. Moreover, in a study that analyzed 242 HCM patients who had never been diagnosed with AF, new AF occurred in 41 patients (17%) during a 4.8 ± 3.7 year follow-up period, with multivariable analysis showing that LA volume (≥37 mL/m^2^; hazard ratio, 2.68; *p* = 0.008) predicted new AF, but LA diameters (≥45 mm; hazard ratio, 1.67; *p* = 0.145) did not significantly predict it, although univariate analysis revealed that both LA volume and LA diameters predicted new AF [20]. 

In clinical practice, when measuring LA volume, maximum LV volume is typically assessed; however, the minimum LA volume (measured at the LV end-diastole) should also be evaluated because the minimum LA volume may better reflect LV end-diastolic pressure because the LA is constantly exposed and affected by LV pressure during diastole [27]. A recent study in HCM patients reported that minimum LA volume demonstrated superior predictive ability for clinical outcomes (heart failure hospitalization, stroke, and all-cause mortality) compared to maximum LA volume [28]. A previous cohort study showed that the minimum LA volume was marginally superior to the maximum LA volume in predicting the occurrence of first AF [29]. Therefore, attention should be paid to the minimum LA volume and the maximum LA volume; notably, LA function can be evaluated using these values. LA size is also useful for predicting therapeutic effects in AF patients. A recent meta-analysis reported that LA volume and LAVI were higher in patients with AF recurrence following AF ablation compared to patients without AF recurrence [19]. 

In addition to LA size, LA function is also an important indicator of clinical progress in patients with AF [30,31]. Importantly, it takes time for the LA to be remodeled due to increased LV pressure, such as diastolic dysfunction, and LA dysfunction can be detected before LA enlargement. TTE is the most frequently used imaging modality to assess LA function. The atrial function was evaluated based on atrial contractility and wall deformation. LA phasic function was used to evaluate atrial contractility using the LA volume for each phase. 

LA function in sinus rhythm can be subdivided into three components: (1) reservoir, (2) conduit function, and (3) booster pump function. Both the passive (conduit) and active (booster pump function) phases are components of atrial emptying in sinus rhythm. However, there is only a passive phase during AF; therefore, the term “ejection fraction” was used instead of “emptying fraction” [27]. The total, passive, and active emptying fractions were calculated using LA volume measured in three different phases (Table 1). There is a limit to the measurement of the LA volume immediately before atrial contraction (LAV_preA_) during AF. However, LA phasic function measured using the volumetric method in TTE is difficult to use in actual clinical practice because of the great variability between individuals. Currently, strain analysis is widely used to assess LA phasic function. Furthermore, LA function can be assessed using tissue Doppler imaging (TDI) or speckle-tracking (ST) measurements of LA strain and strain rate [32]. However, the angle dependency of the method is a critical limitation of TDI-derived strain measurement [33]. If the angle of interrogation exceeds 20 degrees, the velocity could be underestimated. ST measurement have the advantage of angle independency. Notably, previous studies have validated the feasibility and reproducibility of ST echocardiography for evaluating LA function [34,35]. Recently, ST echocardiography has been widely used to quantitatively assess the three phases of LA function. The longitudinal strain curves were generated for each of the twelve LA segments in the four-chamber and two-chamber views, and the reservoir, conduit, and contractile strains were obtained. (Figure 1). Reserve strain is a commonly used global longitudinal LA strain value that reflects atrial function and provides information about atrial fibrosis. Kuppahally et al. showed that mid-lateral strain and strain rate were significantly correlated with LA fibrosis, as assessed using CMR in patients with AF [36]. In patients with HCM, LA strain is strongly associated with AF. Vasques et al. demonstrated that LA reservoir strain (20.7% vs. 24.6%) and conduit strain (10.1% vs. 13.0%) were impaired in HCM patients with paroxysmal AF compared with patients without AF, and low LA reservoir and conduit strain were associated with a higher risk of adverse cardiovascular outcomes in HCM patients with paroxysmal AF [37]. Furthermore, Zegkos et al. demonstrated a significant increase in new-onset AF in HCM patients with 20% or less LA reservoir strain compared with HCM patients with 20% or more LA reservoir strain [38]. Debonnaire et al. reported that LA reserve strain on ST independently predicted new-onset AF in patients with HCM without a previous diagnosis of AF. In addition, LA strain (≤2 3.4%) significantly predicted new AF with LA volume (≥37 mL/m^2^) even in patients with a preserved LA diameter (<45 mm) [20]. In addition, LA function is used for the prediction of AF recurrence after AF ablation. A recent study reported that LA strain contractile function was the independent predictor of AF recurrence after AF ablation [30].

#### 3.1.2. LV Assessment

In HCM, a thickened and noncompliant LV results in impaired diastolic function, which promotes LA remodeling and enlargement. Therefore, LV evaluation is important in patients with HCM. LV evaluation by TTE includes assessment of the location and extent of ventricular hypertrophy, LVOT obstruction, the presence and degree of MR, papillary muscle abnormalities, and the presence of aneurysms. Fundamentally, LV morphology, such as asymmetric septal hypertrophy, should be evaluated in patients with HCM using TTE. Therefore, it is important to evaluate the degree of ventricular hypertrophy and myocardial function, including diastolic and systolic functions. In HCM, accurate measurement of the wall thickness is important. In the parasternal long-axis view, the interventricular septum and posterior walls were measured at the end diastole. Notably, wall thickness is associated with sudden cardiac death and AF in patients with HCM and is a very important parameter in clinical decisions, such as implantable cardiac defibrillator implantation. A previous study has reported that LV wall thickness is associated with AF [39]. However, maximal wall thickness showed high inter-reader variability on TTE [40]. Moreover, there is a limitation in measuring LV mass using TTE because the LV shape is asymmetric in patients with HCM. 

As mentioned earlier, LV diastolic dysfunction serves as a trigger for AF and promotes the formation of a structural substrate called atrial remodeling, contributing to the generation and maintenance of AF. For the assessment of LV diastolic function in HCM patients, current guidelines recommend a comprehensive approach including the E/e’ ratio (>14), LAVI (>34 mL/m^2^), pulmonary vein atrial reversal velocity (Ar-A duration ≥ 30 ms; A, mitral end-diastolic inflow; Ar, pulmonary vein reversal flow), and peak velocity of TR jet by CW Doppler (>2.8 m/s), which can be applied with or without the presence of LVOT obstruction [41]. A recent study of 290 patients with HCM showed that if more than three of the four variables are presented in the guidelines, the risk of HCM-related adverse outcomes increases [42]. During AF, the transmitral A and pulmonary Ar velocities are absent because atrial contraction is lost. The E/e’ ratio is reliable for estimating the LV filling pressure in patients with AF [43]. Furthermore, a previous study reported that the septal E/e ratio predicts adverse outcomes in children with HCM [44]. Finally, Okamatsu et al. reported that E/e’ is an independent predictor of AF recurrence after catheter ablation in patients with HCM [45]. 

Usually, LV systolic function is normal to hyperdynamic in patients with HCM. In HCM, the most basic parameter for LV systolic function is the LV ejection fraction (EF). Currently, it is recommended that the biplane method is optimal for LVEF measurements. LV systolic dysfunction in HCM is defined as LV < 50%, and previous studies have reported that the proportion of LV systolic dysfunction in HCM is between 4% and 9% [46,47,48]. Clinical events, including AF, occur more frequently in patients with HCM with LV systolic dysfunction than in those without LV systolic dysfunction. Marstrand et al. reported that AF events occurred in 49.3% and 20.9% of HCM patients with LV systolic function compared to patients without LV systolic dysfunction from the large international SHaRe Registry (Sarcomeric Human Cardiomyopathy Registry) [48]. However, since HCM is accompanied by myocardial damage caused by myocyte disarray, a ventricular myocardial function cannot be normal, even if it is subclinical. Although LVEF is normal, the indicator that can reflect this in the presence of subclinical ventricular dysfunction is strain. 

Two methods, TDI and ST, are used to evaluate LA. The TDI method is limited to measuring movement parallel to the ultrasound beam, whereas the ST method can assess movement independent of the angle. Figure 2 shows representative cases of strain measurements using the ST method. The LV global longitudinal strain (GLS) reflects subclinical myocardial impairment, especially in the setting of normal LVEF, which is consistent in patients with HCM [49,50,51]. 

Vasquez et al. reported that LV GLS is impaired in HCM patients with paroxysmal AF compared to patients without AF (13.0 ± 3.3% vs. 15.6 ± 3.0%) [37]. It has also been reported that a lower GLS value is related to new-onset AF, and one study showed a significant increase in new-onset AF in HCM patients with −14% or less GLS compared to HCM patients with −14% or more GLS [38]. 

Notably, analyzing regional wall mechanics with strain is helpful. Segments with normal thickness but reduced strain values are occasionally observed, and such segments may progress with myocardial disarray and fibrosis even in the absence of an increase in thickness. Urtado et al. showed that the regional longitudinal segment strain value decreased in segments with significant fibrosis evaluated using CMR, even after adjusting the wall thickness [52]. 

In addition, LVOT obstruction, systolic anterior motion of the mitral leaflet, and MR findings, which increase the risk of AF, must be evaluated using TTE. 

A large prospective observational trial reported that moderate or severe MR was one of the predictors of major AF events, defined as those requiring electrical cardioversion, catheter ablation, hospitalization for >24 h, or clinical decisions to accept permanent AF [53]. 

### 3.2. CMR

Current guidelines recommend conducting CMR imaging when it is unclear whether to diagnose HCM using echocardiography or when it is required to differentiate it from other infectious diseases [54]. CMR imaging is the gold standard for cardiac chamber evaluation and can also be used to evaluate LV wall thickness and function regardless of the presence of an aneurysm. In addition, because tissue characterization is possible with CMR, analysis of LV fibrosis can be performed, which enables the measurement of the degree of myocardial fibrosis. LA function, size, and fibrosis can also be evaluated, providing additional information in addition to echocardiography for analyzing the LA in patients with AF. 

#### 3.2.1. LA assessment

Although TTE is the most widely used modality for measuring LA size and LA phasic function, CMR is considered an emerging technology because of its high-quality images, acquisition at a high spatial resolution, wide field of view, high tracking quality, high reproducibility, and low inter- and intra-observer variability [55]. Because CMR can more accurately visualize the LA wall border, LA volumes and LA phasic function can be accurately evaluated using CMR. In addition, feature tracking (FT)-CMR is considered a more accurate modality than ST-echocardiography for measuring myocardial strain (Figure 3) [55,56]. 

In a study that continuously monitored the occurrence of AF by inserting ILR into 203 stroke patients who were not diagnosed with AF, it was shown that both LA size and LA phasic function measured by CMR- and CMR-FT-derived LA strain values significantly predicted new AF: LAmax, per 10 mL/m^2^ increase (hazard ratio, 1.25; *p* = 0.009), LAmin per 10 mL/m^2^ increase (hazard ratio, 1.51; *p* = 0.0001), LA total emptying fraction per 5% decrease (hazard ratio, 1.49; *p* < 0.001), LA active emptying fraction per 5% decrease (hazard ratio, 1.46; *p* < 0.001), LA reservoir strain per 5% decrease (hazard ratio, 1.23, *p* = 0.008), LA active strain per 5% decrease (hazard ratio, 1.56; *p* = 0.002), and LA strain rate per 1/s increase (hazard ratio, 2.06; *p* = 0.002) [57]. Furthermore, in a study on patients with HCM, LA analysis using CMR also showed significant results. Specifically, a large prospective observational trial, the HCMR (Hypertrophic Cardiomyopathy Registry) trial, revealed that the CMR-derived LAVI and LA contractile percentages predicted major AF events [53]. Yang et al. analyzed FT-CMR images of HCM patients with normal LA size and showed that LA reservoir and conduit strains were lower in HCM patients than in healthy controls, indicating that LA strain was impaired before LA remodeling occurred. In patients with HCM, LA strain is a more sensitive predictor of atrial involvement than LA size [58]. CMR-FT-derived LA strain also be used in predicting therapeutic effects in AF patients after AF ablation. Habibi et al. reported that CMR-FT-derived peak LA strain was independently associated with AF recurrence after pulmonary vein isolation in AF patients [59]. 

Despite CMR’s advantages of analyzing the degree of myocardial fibrosis owing to the possibility of tissue characterization, assessing atrial fibrosis using LGE is challenging because the LA wall is thin. A previous study that analyzed 65 patients with paroxysmal or persistent AF showed that LA LGE was inversely correlated with LA strain and strain rate, and the extent of LA LGE was greater in patients with persistent AF than in those with paroxysmal AF [36]. A recent study reported that (1) all HCM patients had evidence of LA LGE and (2) the LA LGE amount was higher in the HCM in the AF group than in the no AF group [60]. Interestingly, there was a positive correlation between LA LGE and LV LGE, which suggests that myopathy progresses in both the atrium and ventricle of HCM patients [60]. LA fibrosis is also a critical factor for predicting therapeutic effects in AF patients. Roh et al. analyzed 173 AF patients referred for AF ablation and revealed that freedom from recurrence of atrial arrhythmia occurred in 81% of the small LGE (<20%) group and 55% of the extensive LGE (>20%) group (log rank *p* = 0.014) in patients with non-paroxysmal AF (n = 72) [61]. A recent meta-analysis reported that LA fibrosis quantified by CMR before AF ablation had a significant prognostic value in predicting the risk of AF recurrence after AF ablation. The authors resulted that for every 10% increase in LA fibrosis on CMR, there was a 1.54-fold increase in AF recurrence after AF ablation [62]. 

#### 3.2.2. LV Assessment

In HCM, CMR plays various roles in LV assessment. TTE has several limitations, owing to its poor acoustic windows. CMR can accurately define and differentiate the myocardium. Compared to TTE, CMR enables more accurate measurement of LV wall thickness, LV mass, and systolic function. Because it is sometimes difficult to accurately exclude RV trabeculation or papillary muscle, TTE often overestimates the wall thickness. In addition to being able to accurately evaluate ventricular size and function by visualizing the endocardial border more clearly, CMR can measure strain using FT techniques (Figure 4). Similar to the strain obtained by echocardiography, LV strain values measured by CMR-FT reflected the degree of subclinical myocardial damage in patients with HCM with normal LVEF. Cavus et al. showed that LV longitudinal, circumferential, and radial strain values correlated with NT-proBNP and troponin T values in a study that analyzed 144 patients with HCM with normal LVEF [63]. 

CMR imaging can provide information on myocardial tissue characterization. In particular, LGE is crucial for assessing myocardial fibrosis; after gadolinium administration, it is redistributed from the intravascular space to the extracellular space and accumulates in areas with an expansion of the extracellular space due to myocardial edema or fibrosis, resulting in increased signal intensity on T1-weighted imaging. T1 mapping may provide an advantage over LGE by enabling a more valid quantification of diffuse fibrosis. The native T1 value is a tissue-specific time constant measured without a contrast agent. The T1 relaxation time is measured for each pixel of the myocardium. Fibrotic tissues alter the water composition and T1 values [64]. The extracellular volume fraction (ECV) is calculated by comparing native T1 and post-contrast T1 images. 

Myocardial fibrosis is a hallmark of HCM that contributes to diastolic dysfunction, resulting in LA remodeling, which acts as a substrate for AF. Furthermore, ventricular fibrosis is associated with adverse outcomes, including ventricular arrhythmia, SCD, and death, in HCM [65]. In HCM, myocardial fibrosis is thought to be triggered by premature myocyte death caused by stress imposed directly by sarcomere mutations, a unique characteristic of HCM as a genetic disease. Studies have reported that patients with HCM and sarcomere mutations have a greater fibrotic burden than those without mutations. Vullaganti et al. demonstrated that patients with HCM with sarcomere mutations have greater LGE than patients with HCM without a sarcomere mutation do [66]. Recently, we also reported that patients with sarcomere mutations had a higher prevalence of LV LGE (90% vs. 60%; *p* < 0.001), burden of LV LGE (10.6 ± 10.1 vs. 6.4 ± 9.3%, *p* = 0.040), LGE-involved segments (4.9 ± 2.8 vs. 2.9 ± 3.5; *p* = 0.002), and ECV (34.2 ± 4.8% vs. 31.4 ± 4.3%; *p* = 0.001) [67]. The relationship between ventricular fibrosis and ventricular arrhythmia is known, but its relationship with atrial arrhythmia remains unclear. A previous study that analyzed autopsied hearts with HCM showed that LV fibrosis was more severe in patients with HCM and AF than in those without AF [68]. A factor influencing the occurrence of AF by myocardial fibrosis is that it is linked to diastolic dysfunction, and our previous study reported that the LV LGE segment number was associated with Doppler-derived E/e’ [11]. Notably, previous reports demonstrated that LGE is associated with the occurrence of AF in patients with HCM [69,70]. Guo Y et al. reported that transforming growth factor–beta (TGF-β), a profibrotic cytokine that promotes cardiac fibrosis, independently predicted postoperative AF in HCM after surgical ventricular septal myectomy [71]. Furthermore, Tian et al. reported that the LV remodeling index was higher in the AF group than in the non-AF group of patients with HCM [72]. However, LV LGE was less associated with AF than LA size, and multivariate analysis showed that LA volume and diastolic dysfunction were associated with AF in patients with HCM, whereas LV LGE was not significantly associated with AF in patients with HCM [69]. In addition, LV LGE did not show significant predictive power in a previous study that reported that LA contractility and LA volume predicted the occurrence of AF [53]. 

## 4. The Genetic Influence on Atrial Myopathy and AF in Patients with HCM

As previously discussed, factors related to atrium and ventricle size and function, as well as fibrotic burden, are associated with the occurrence of AF in patients with HCM. However, in addition to these factors, the unique characteristics of HCM as a genetic disease must be considered. Sarcomere gene mutations can lead to dysregulation of calcium handling, thereby increasing arrhythmogenicity [73]. Although the exact extent to which genetic factors contribute to atrial myopathy remains unclear, recent studies have suggested that genetic mutations may play a role in atrial myopathy [11,74]. Wang et al. analyzed 105 HCM patients who underwent genetic testing for eight HCM-associated sarcomere genes (myosin-binding protein C (MYBPC3), β-myosin heavy chain (MYH7), essential and regulatory myosin light chains, cardiac troponin T, cardiac troponin I, α-tropomyosin, and cardiac actin) and three HCM phenocopy genes (LAMP2 for Danon disease, GLA for Fabry disease, and PRKAG2 for PRKAG2 cardiomyopathy) along with CMR. They reported that in the mutation-positive group, LAVI was higher, and LA reservoir strain and booster-pump strain were lower compared to the mutation-negative group [74]. In our study analyzing 135 patients who underwent both CMR and genetic testing, we reported a significant reduction in the LA total emptying fraction in the pathogenic or likely pathogenic sarcomere gene mutation group compared to the mutation-negative group. In this study, the LA total emptying fraction was found to be independent of LV LGE, indicating that LA myopathy in HCM is associated with sarcomere gene mutations independently of LA loading conditions [11]. Although LA remodeling and dysfunction may contribute to the onset and maintenance of AF, in HCM, sarcomere gene mutations may also be directly associated with AF. Lee et al. demonstrated in the multicenter registry The Sarcomeric Human Cardiomyopathy Registry (SHaRe) that among patients with coexisting MYH7, MYBPC3, and thin filament mutations, the MYH7 mutation group showed the highest incidence of AF, even after adjusting for LA size.

## 5. Conclusions

HCM is a disease with diverse pathophysiology and ventricular hypertrophy. TTE is a fundamental tool for assessing various hemodynamic phenomena and remains pivotal in imaging. However, TTE images have inherent limitations, whereas CMR offers a more precise assessment of cardiac morphology, chamber dimensions, and function. AF is a common arrhythmia in HCM, and its co-occurrence in HCM can significantly impact clinical outcomes. Therefore, early detection and management of AF are crucial in HCM patients. TTE or CMR can be utilized to assess the extent of atrial and ventricular remodeling as well as their function in HCM patients, serving as predictors for the development of new AF. Using such imaging modalities, it would be possible to detect AF early in HCM patients at risk of new AF through more intensive electrocardiography monitoring and surveillance. These measures can also be employed as indicators for prognostication in HCM patients with AF undergoing treatment. 

Furthermore, CMR assumes significance as it enables the evaluation of myocardial fibrosis, which is directly linked to adverse outcomes in HCM patients. Therefore, a comprehensive diagnosis and treatment approach for HCM entails evaluating myocardial function from multiple perspectives and measuring fibrotic burden using varied imaging techniques. Further research and data accumulation are necessary, particularly in domains such as LA fibrosis and strain assessment through FT-CMR.

## Figures and Tables

**Figure 1 diagnostics-13-03049-f001:**
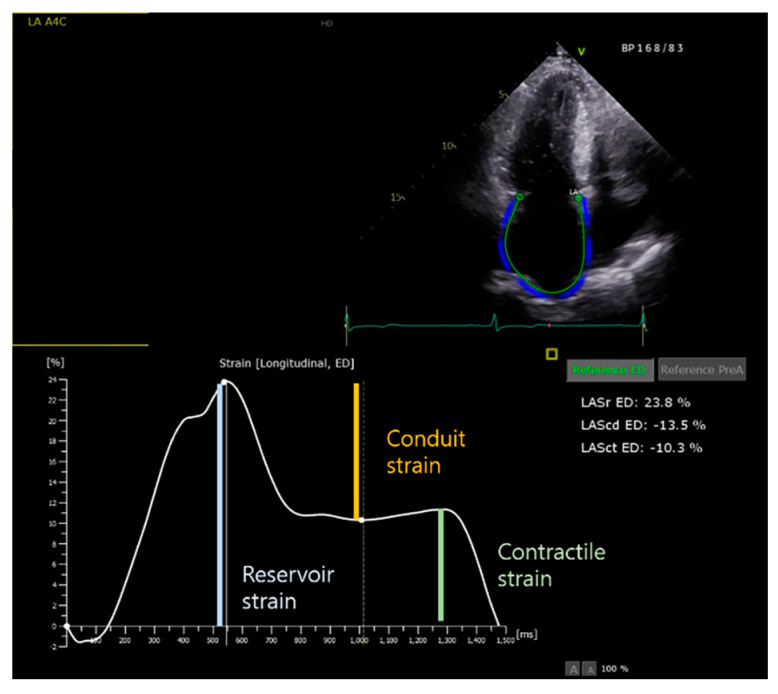
Speckle-tracking analysis and strain measurements of phasic LA strain. LAScd, left atrial conduit strain; LASct, left atrial contractile strain; and LASr, left atrial reservoir strain.

**Figure 2 diagnostics-13-03049-f002:**
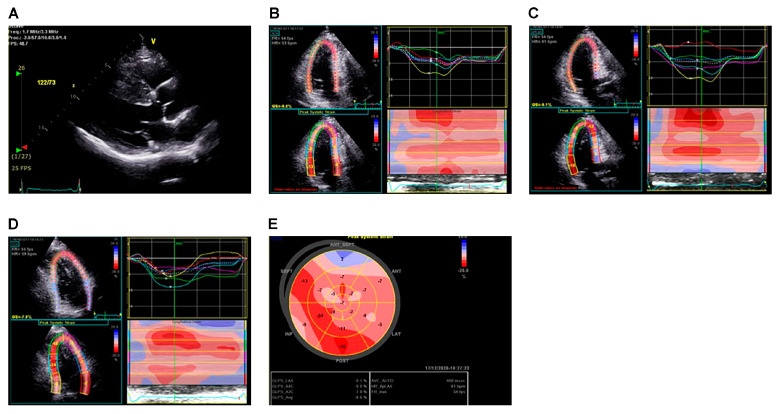
A representative left ventricular strain image by two-dimensional speckle-tracking echocardiography in an HCM patient with asymmetrical septal hypertrophy. Two-dimensional echocardiographic image of asymmetrical septal hypertrophy (**A**). Global and regional longitudinal deformation assessed by speckle-tracking analysis from apical 4 chamber (**B**), 3 chamber (**C**) and 2 chamber (**D**) views. The global longitudinal strain (GLS) is –8.6% (**E**). HCM, hypertrophic cardiomyopathy.

**Figure 3 diagnostics-13-03049-f003:**
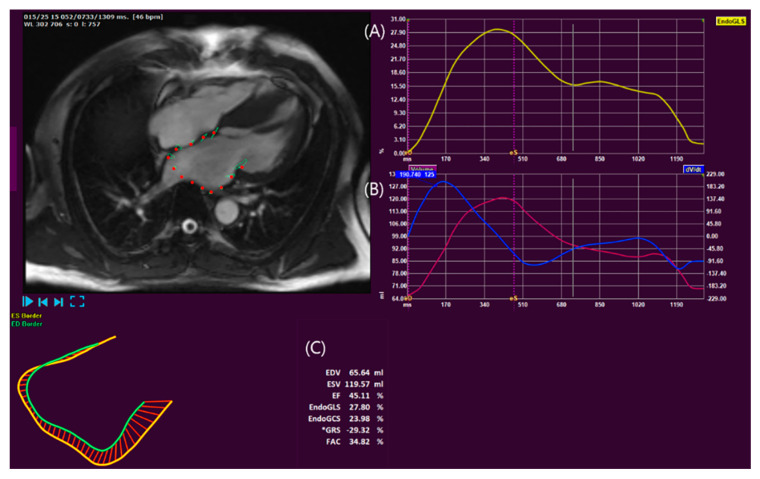
A representative left atrial strain image by CMR-FT in HCM patients. Left atrial strain (**A**) and volume (**B**). The ejection fraction is 45%, and the global longitudinal strain is 27.8% (**C**). CMR, cardiac magnetic resonance; FT, feature tracking.

**Figure 4 diagnostics-13-03049-f004:**
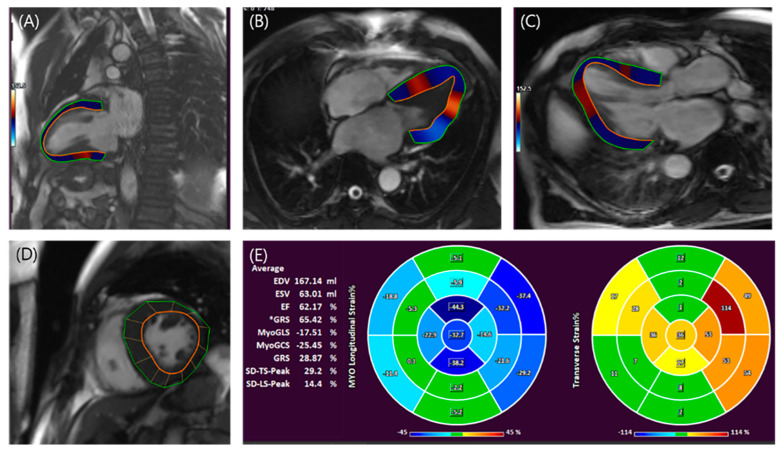
A representative left ventricular strain imaging by CMR-FT in an HCM patient. LV endocardial and epicardial contours are manually delineated at the end-diastole phase in 2 chamber (**A**), 4 chamber (**B**), LV outflow tract (**C**), and sort axis (**D**) cine images. The global longitudinal strain is −17.51% (**E**). CMR, cardiac magnetic resonance; FT, feature tracking; LV, left ventricle.

**Table 1 diagnostics-13-03049-t001:** LA phasic functions.

LA total emptying fraction, %	(LAV_max_ − LAV_min_) / LAV_max_
LA reserve fraction, %	(LAV_max_ − LAV_min_) / LAV_min_
LA passive emptying (conduit) fraction, %	(LAV_max_ − LAV_preA_) / LAV_max_
LA active emptying (booster pump) fraction, %	(LAV_preA_ − LAV_min_) / LAV_pre-A_

LAV, left atrial volume; LAV_max_, maximal left atrial volume; LAV_min_, minimal left atrial volume; LAV_pre-A_, preA left atrial volume.

## Data Availability

Not applicable.

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
