# Peer review of "Multimodality Imaging in Patients with Hypertrophic Cardiomyopathy and Atrial Fibrillation"

_diagnostics, 2023, doi:10.3390/diagnostics13193049_

Round 1

Reviewer 1 Report

Comments and Suggestions for Authors

The present paper aimed to review the role of multimodality imaging in Hypertrophic Cardiomyopathy (HCM) patients with Atrial Fibrillation (AF).

A few changes are needed, as follows:

Please explain every abbreviation before using it.

Introduction: Please also mention oxidative stress related to HCM and AF (Szyguła-Jurkiewicz B, et al.  Oxidative Stress Markers in Hypertrophic Cardiomyopathy. Medicina (Kaunas). 2021 Dec 24;58(1):31. doi: 10.3390/medicina58010031 and Gasparova I, et al.  Perspectives and challenges of antioxidant therapy for atrial fibrillation. Naunyn Schmiedebergs Arch Pharmacol. 2017 Jan;390(1):1-14. doi: 10.1007/s00210-016-1320-9).

Line 124-126: you state: “TDI method has some limitations, which including angle-dependent and susceptible to reverberations, sidelobes and drop out artifacts.” Please provide more information about “angle-dependent and susceptible to reverberations, sidelobes and drop out artifacts.”

Please add a few words about the relationship HCM-AF-liver cirrhosis. Zamirian et al (Zamirian M, et al. Liver cirrhosis prevents atrial fibrillation: A reality or just an illusion? J Cardiovasc Dis Res. 2012;3:109–112) suggested that liver cirrhosis has a protective effect against atrial fibrillation, despite significant metabolic abnormalities, inflammatory syndrome and enlarged left atria. The low prevalence of atrial fibrillation may be the result of the accumulation of anti-arrhythmic or anti-inflammatory substances that are normally metabolized by an intact functioning liver; this would explain the development of atrial fibrillation after liver transplantation. The administration of spironolactone and beta-blockers is also able to reduce atrial excitability.

Please mention that after liver transplantation, in patients with preoperative diagnosis of HCM, preoperative left atrium size (measured by echocardiography) and the sudden cardiac death risk were not significantly associated with 1-year survival (Pai SL, et al. Hypertrophic Cardiomyopathy in Liver Transplantation Patients. Transplant Proc. 2018 Jun;50(5):1466-1469. doi: 10.1016/j.transproceed.2018.02.080).

Author Response

The present paper aimed to review the role of multimodality imaging in Hypertrophic Cardiomyopathy (HCM) patients with Atrial Fibrillation (AF).

A few changes are needed, as follows:

Please explain every abbreviation before using it.

Introduction: Please also mention oxidative stress related to HCM and AF (Szyguła-Jurkiewicz B, et al.  Oxidative Stress Markers in Hypertrophic Cardiomyopathy. Medicina (Kaunas). 2021 Dec 24;58(1):31. doi: 10.3390/medicina58010031 and Gasparova I, et al.  Perspectives and challenges of antioxidant therapy for atrial fibrillation. Naunyn Schmiedebergs Arch Pharmacol. 2017 Jan;390(1):1-14. doi: 10.1007/s00210-016-1320-9).

->Thank you for your valuable comments. I added the comments about contribution of oxidative stress to HCM and AF as follows: “Although primary impairments of sarcomeres caused by sarcomere gene variants are associated with the progression of LV remodeling, but increased oxidative stress can also contribute to secondary alterations of sarcomeres. Oxidative stress strengthens sarcomere mutation, and mutant sarcomere proteins becomes a source of reactive oxygen species and forms a cycle that worsens each other.(14) Oxidative stress results in inflammatory changes and contribute to altered ion channel activity by various pathways, and AF may activate different oxidative pathways resulting in increasing oxidative stress.(15) Oxidative stress can contribute to both the progress of HCM and the occurrence and maintenance of AF by affecting cardiac remodeling.” (line 69-78) with a citation of recommended references.

Line 124-126: you state: “TDI method has some limitations, which including angle-dependent and susceptible to reverberations, sidelobes and drop out artifacts.” Please provide more information about “angle-dependent and susceptible to reverberations, sidelobes and drop out artifacts.”

->Thank you for your valuable comments. There was a problem with our expression. I added contents about this as follows: “However, the angle dependency of the method is a critical limitation of TDI-derived strain measurement. (32) If the angle of interrogation exceeds 20 degrees, the velocity could be underestimated. ST measurement have the advantage of angle independency.” (line 61-64) Thank you.

Please add a few words about the relationship HCM-AF-liver cirrhosis. Zamirian et al (Zamirian M, et al. Liver cirrhosis prevents atrial fibrillation: A reality or just an illusion? J Cardiovasc Dis Res. 2012;3:109–112) suggested that liver cirrhosis has a protective effect against atrial fibrillation, despite significant metabolic abnormalities, inflammatory syndrome and enlarged left atria. The low prevalence of atrial fibrillation may be the result of the accumulation of anti-arrhythmic or anti-inflammatory substances that are normally metabolized by an intact functioning liver; this would explain the development of atrial fibrillation after liver transplantation. The administration of spironolactone and beta-blockers is also able to reduce atrial excitability.

->Thank you for your recommendation about very interesting observations. According to your recommendation, we added the following comments with citation of your recommended references as follows with citation of reference: “One of the interesting observations is that liver cirrhosis has a protective effect against AF, despite significant metabolic abnormalities, inflammatory syndrome and enlarged LA. The low prevalence of atrial fibrillation (AF) may be the result of the accumulation of anti-arrhythmic or anti-inflammatory substances that are normally metabolized by an intact functioning liver; this would explain the development of AF after liver transplantation. The administration of spironolactone and beta-blockers is also able to reduce atrial excitability.” (line 81-87). Thank you.

Please mention that after liver transplantation, in patients with preoperative diagnosis of HCM, preoperative left atrium size (measured by echocardiography) and the sudden cardiac death risk were not significantly associated with 1-year survival (Pai SL, et al. Hypertrophic Cardiomyopathy in Liver Transplantation Patients. Transplant Proc. 2018 Jun;50(5):1466-1469. doi: 10.1016/j.transproceed.2018.02.080).

->Thank you for your comment and recommendation. We added your recommended comments as follows with citation of reference: “In addition, in liver transplantation patients with preoperative diagnosis of HCM, preoperative LA size (measured by echocardiography) and the sudden cardiac death risk were not significantly associated with 1-year survival.” (line 87-90) Thank you.

Reviewer 2 Report

Comments and Suggestions for Authors

 In the presented manuscript Hyemoon Chung and Eui-Young Choi aimed to review literature to establish whether  echocardiography and cardiac magnetic resonance based left atrial and ventricular assessment can be useful among hypertrophic  cardiomyopathy  population  with atrial fibrillation.  The authors should be acknowledged for their effort put into the manuscript. However there are many caveats that should be corrected.

1.       The aims of review should be more precise. The authors state that “this review is to focus on what should be evaluated using multimodality imaging including echocardiography and cardiac magnetic resonance (CMR), especially  in HCM patients with AF (line 37-39)”. Actually the review focuses on AF prediction and disease prognosis in HCM population on the basis of LA and LV assessment with echo and cMR. Moreover change/delete expressions such as “especially in HCM patients in AF” (review concerns HCM population exclusively) and “what should be evaluated using multimodality imaging” (the authors do no evaluate anything)

2.      Echocardiography and cMR based LA and LV assessment: The authors intensively describe general, widely accepted methods for measurement LA and LV size and function. This should be deleted (text and figures), as there is no novelty behind this. The authors should discuss which parameters are useful or not in the context of HCM and AF exclusively. Otherwise it seems to be an imaging textbook.

3.      Conclusions [line 340-350]: Very badly written. There is no information how echo and cMR could help in  the diagnosis and  management of AF in HCM population

4.      The manuscript should include information how echo and cMR could help in invasive AF management, especially AF ablation  

5.      Many speculations instead of well accepted facts could be found along the manuscript. It especially concerns pathophysiology of AF. A complex interplay of triggers, perpetuators, and substrate development eventually result in AF occurrence. Various factors cause complex atrial alterations, including stretch-induced fibrosis, hypocontractility, fatty infiltration, inflammation, vascular remodelling, ischaemia, ion-channel dysfunction, and calcium instability. All enhance ectopy and conduction disturbances, increase atrial propensity to develop/maintain AF, and facilitate the AF-associated hypercoagulable state. AF in itself aggravates many of these mechanisms, which may explain its progressive nature.

a)      Diastolic dysfunction and atrial remodeling contribute to the occurrence of atrial fibrillation (AF) (line 30-32)- this is speculation, provide  reference

b)     Generally, LA remodeling is a major risk factor of AF (line 45) - this is speculation, provide  reference

c)      Therefore, evaluating the degree of LA remodeling and LA function when AF is accompanied in HCM patients is the most important for cardiac imaging evaluation (line 52-53) - this is speculation, provide  reference

d)     An accurate information of LA size and function provides crucial information to the clinician for AF prognosis and treatment (line 68-69) - this is speculation, provide  reference

e)      Furthermore, this information can also be used to predict future AF events in HCM patients whose AF has not yet been diagnosed (line 69-71) - provide reference

f)       Accurate determination of LA size is important for patient risk stratification. LA size is a key predictor of future AF development, and clinical outcome including stroke among patients with AF (provided reference Osranek M, et al. Left atrial volume predicts cardiovascular events in patients originally diagnosed with lone atrial fibrillation: three-decade follow-up. Eur Heart J. 2005) [line 77-79] - this is based on very old study and is not in line with recent guidelines

g)      In addition to LA size, LA function is also an important indicator in predicting clinical progress in AF patients [line 103-104] - this is speculation, provide  reference

6.      Myocyte disarray and myofibril disruption, and increased interstitial space with fibrosis results in diastolic dysfunction and also provides a substrate for supraventricular and ventricular arrhythmia. Diastolic dysfunction and atrial remodeling contribute to the occurrence of atrial fibrillation (AF), so AF is a frequent supraventricular arrhythmia in patients with HCM.  In HCM, especially, ventricular fibrotic burden can contribute to AF generation by greatly influencing diastolic and atrial functions, so evaluation for ventricles as well as atrium is important. (line 28-36)- specify if diastolic dysfunction concerns LA or LV

Author Response

 In the presented manuscript Hyemoon Chung and Eui-Young Choi aimed to review literature to establish whether echocardiography and cardiac magnetic resonance based left atrial and ventricular assessment can be useful among hypertrophic cardiomyopathy population with atrial fibrillation.  The authors should be acknowledged for their effort put into the manuscript. However there are many caveats that should be corrected.

  1. The aims of review should be more precise. The authors state that “this review is to focus on what should be evaluated using multimodality imaging including echocardiography and cardiac magnetic resonance (CMR), especially in HCM patients with AF (line 37-39)”. Actually the review focuses on AF prediction and disease prognosis in HCM population on the basis of LA and LV assessment with echo and cMR. Moreover change/delete expressions such as “especially in HCM patients in AF” (review concerns HCM population exclusively) and “what should be evaluated using multimodality imaging” (the authors do no evaluate anything)

->Thank you for your critical pointing out. According to your recommendation, we revised the paragraph as follows by emphasizing this point as follows: “This review aimed to focus on what should be evaluated using multimodal imaging, including echocardiography and cardiac magnetic resonance (CMR) in terms of AF prediction and disease prognostication in HCM population on the basis of LA and LV assessment.” (line 46-49) We also deleted the phrase of “HCM patients with AF”

  1. Echocardiography and cMR based LA and LV assessment: The authors intensively describe general, widely accepted methods for measurement LA and LV size and function. This should be deleted (text and figures), as there is no novelty behind this. The authors should discuss which parameters are useful or not in the context of HCM and AF exclusively. Otherwise it seems to be an imaging textbook.

->Thank you for your advice. We specified the role of echo and CMR based LA and LV functional and structural assessment, instead of general concept. We deleted the contents about general concept, and revised the text. We also deleted one figure (Figure 4).

  1. Conclusions [line 340-350]: Very badly written. There is no information how echo and cMR could help in the diagnosis and management of AF in HCM population

->Thank you for your comment. We added the context about information how TTE and CMR could help in the diagnosis and management of AF in HCM population as follows according to your recommendation: “AF is a common arrhythmia in HCM and its co-occurrence in HCM can significantly impact clinical outcomes. Therefore, early detection and management of AF are crucial in HCM patients. TTE or CMR can be utilized to assess the extent of atrial and ventricular remodelling as well as their function in HCM patients, serving as predictors for the development of new AF. Using such imaging modalities, it would be possible to detect AF early in HCM patients at risk of new AF through more intensive electrocardiography monitoring and surveillance. These measures can also be employed as indicators for prognostication in HCM patients with AF undergoing treatment“ (line 452-460).

  1. The manuscript should include information how echo and cMR could help in invasive AF management, especially AF ablation  

-> Thank you for your crucial comment. We added contents about imaging parameters which predict success of AF ablation in HCM. (line 38-40, 86-88, 325-329, 340-349).

  1. Many speculations instead of well accepted facts could be found along the manuscript. It especially concerns pathophysiology of AF. A complex interplay of triggers, perpetuators, and substrate development eventually result in AF occurrence. Various factors cause complex atrial alterations, including stretch-induced fibrosis, hypocontractility, fatty infiltration, inflammation, vascular remodelling, ischaemia, ion-channel dysfunction, and calcium instability. All enhance ectopy and conduction disturbances, increase atrial propensity to develop/maintain AF, and facilitate the AF-associated hypercoagulable state. AF in itself aggravates many of these mechanisms, which may explain its progressive nature.  

a) Diastolic dysfunction and atrial remodeling contribute to the occurrence of atrial fibrillation (AF) (line 30-32)- this is speculation, provide  reference

->Thank you for your comment. We added the reference at line 38.

b)     Generally, LA remodeling is a major risk factor of AF (line 45) - this is speculation, provide reference,

-> Thank you for your comment. We revised the sentence and added the references at line 55.

c)      Therefore, evaluating the degree of LA remodeling and LA function when AF is accompanied in HCM patients is the most important for cardiac imaging evaluation (line 52-53) - this is speculation, provide reference.

-> Thank you for your comment. We agree that the expression “most important” is subjective, therefore we revised the sentence. This sentence if based on the content described before this sentence in this paragraph, and the references of the previous content and the preceding sentences corresponds to the references to this sentence. We added the references at line 65.

d)     An accurate information of LA size and function provides crucial information to the clinician for AF prognosis and treatment (line 68-69) - this is speculation, provide reference.

-> Thank you for your comment. We added the reference at line 100-101.

e)      Furthermore, this information can also be used to predict future AF events in HCM patients whose AF has not yet been diagnosed (line 69-71) - provide reference.

-> Thank you for your comment. We added the reference at line 102.

f)       Accurate determination of LA size is important for patient risk stratification. LA size is a key predictor of future AF development, and clinical outcome including stroke among patients with AF (provided reference Osranek M, et al. Left atrial volume predicts cardiovascular events in patients originally diagnosed with lone atrial fibrillation: three-decade follow-up. Eur Heart J. 2005) [line 77-79] - this is based on very old study and is not in line with recent guidelines

-> Thank you for your comment. We deleted the reference because the reference is old and applied to general population not specific for HCM. Instead, we cited recent references regarding risk stratification with LA size in HCM for the development of AF and prognosis in AF patients at line 111.

g)      In addition to LA size, LA function is also an important indicator in predicting clinical progress in AF patients [line 103-104] - this is speculation, provide reference.

-> Thank you for your comment. We added the reference at line 42.

  1. Myocyte disarray and myofibril disruption, and increased interstitial space with fibrosis results in LV diastolic dysfunction and also provides a substrate for supraventricular and ventricular arrhythmia. The LV Diastolic dysfunction and corresponding atrial remodeling contribute to the occurrence of atrial fibrillation (AF), so AF is a frequent supraventricular arrhythmia in patients with HCM.  In HCM, especially, ventricular fibrotic burden can contribute to AF generation by greatly influencing LV diastolic and atrial functions, in addition recent studies showed that inborn atrial myopathy also contribute to development of AF. Therefore.so evaluation for ventricles as well as atrium is important. (line 28-36)- specify if diastolic dysfunction concerns LA or LV.

->Thank you for your comment. In this paragraph, there is an insufficient clarification of the terms ‘atrium’ and ‘ventricle’. Instead of the concept of ‘diastolic function’, it has been described more precisely as left ventricular (LV) diastolic dysfunction’, and the paragraph has been revised accordingly (line 34-44). Thank you.

Reviewer 3 Report

Comments and Suggestions for Authors

Manuscript is well written and described. I have a question.

I think you should touch on genetic abnormalities in HCM and the status of AF and its relationship to Imaging. 

Author Response

I think you should touch on genetic abnormalities in HCM and the status of AF and its relationship to Imaging. 

->Thank you for your valuable comment. We have incorporated content regarding the correlation between AF and genetic attributes, as well as findings from imaging, in HCM patients as follows: “4. The genetic influence on atrial myopathy and AF in patients with HCM

As previously discussed, factors related to atrium and ventricle size and function, as well as fibrotic burden, are associated with the occurrence of AF in patients with HCM. However, in addition to these factors, the unique characteristics of HCM as a genetic disease, must be considered. Sarcomere gene mutations can lead to dysregulation of calcium handling, thereby increasing arrhythmogenicity.(72) While the exact extent to which genetic factors contribute to atrial myopathy remains unclear, recent studies have suggested that genetic mutations may play a role in atrial myopathy.(11, 73) Wang et al. analyzed 105 HCM patients who underwent genetic testing for eight HCM-associated sarcomere genes [myosin-binding protein C (MYBPC3), β-myosin heavy chain (MYH7), essential and regulatory myosin light chains, cardiac troponin T, cardiac troponin I, α-tropomyosin, and cardiac actin] and three HCM phenocopy genes (LAMP2 for Danon disease, GLA for Fabry disease, and PRKAG2 for PRKAG2 Cardiomyopathy) along with CMR. They reported that in the mutation-positive group, LAVI was higher, and LA reservoir strain and booster-pump strain were lower compared to the mutation-negative group.(73) In our study, analyzing 135 patients who underwent both CMR and genetic testing, we reported a significant reduction in LA total emptying fraction in the pathogenic or likely pathogenic sarcomere gene mutation group compared to the mutation-negative group. In this study, LA total emptying fraction was found to be independent of LV LGE, indicating that LA myopathy in HCM is associated with sarcomere gene mutations independently of LA loading conditions.(11) While LA remodeling and dysfunction may contribute to the onset and maintenance of AF, in HCM, sarcomere gene mutations may also be directly associated with AF. Lee et al. demonstrated in the multicenter registry, The Sarcomeric Human Cardiomyopathy Registry (SHaRe), that among patients with coexisting MYH7, MYBPC3, and thin filament mutations, the MYH7 mutation group showed the highest incidence of AF, even after adjusting for LA size.” (line 418-446). Thank you.

Round 2

Reviewer 2 Report

Comments and Suggestions for Authors

It seems that  the revised version of manuscript is significantly improved. The authors included  all recommended suggestions . However there are still  some minor issues to sort out.

Line 129-133 We mainly measured the maximum LA volume (measured at LV end systole) as the LA volume value; however, the minimum LA volume (measured at LV end-diastole) should also be evaluated because the minimum LA volume may better reflect LV end-diastolic pressure because the LA is constantly exposed and affected by LV pressure during diastole. [27 Thomas L, Marwick TH, Popescu BA, Donal E, Badano LP. Left Atrial Structure and Function, and Left Ventricular Diastolic Dysfunction: JACC State-of-the-Art Review. Journal of the American College of Cardiology. 2019;73(15):1961-77.]

 This sentence suggests that the authors evaluated  LA volume systematically . However it is not followed by any proper reference.  Moreover the authors cite other paper instead.

Line 138-140 A recent meta-analysis reported that LAV and LAVI were higher in patients with AF recurrence following AF ablation compared to patients without AF recurrence.(19)

Abbreviation LAV (LA volume, I guess) has not been explained before.  

Author Response

It seems that the revised version of manuscript is significantly improved. The authors included all recommended suggestions . However there are still  some minor issues to sort out.

Line 129-133 We mainly measured the maximum LA volume (measured at LV end systole) as the LA volume value; however, the minimum LA volume (measured at LV end-diastole) should also be evaluated because the minimum LA volume may better reflect LV end-diastolic pressure because the LA is constantly exposed and affected by LV pressure during diastole. [27 Thomas L, Marwick TH, Popescu BA, Donal E, Badano LP. Left Atrial Structure and Function, and Left Ventricular Diastolic Dysfunction: JACC State-of-the-Art Review. Journal of the American College of Cardiology. 2019;73(15):1961-77.]

 This sentence suggests that the authors evaluated LA volume systematically . However it is not followed by any proper reference. Moreover the authors cite other paper instead.

=>Response: Thank you for your valuable comment. Indeed, measuring minimal LA volume is considered important both theoretically and based on various studies. Our research group has also measured minimal LA volume using CMR for research analysis. However, like many other institutions, in the clinical setting, we primarily measure maximal LA volume due to practical constraints such as time limitations during echocardiography examinations. We do plan to routinely measure minimal LA volume in the future. The minimal LA volume we presented in our paper was obtained using CMR, not echocardiography. Therefore, in the sentence discussing echocardiography, we did not reference our own paper. I am concerned that the wording of this sentence might inadvertently convey that “our group” measured minimal LA volume. I have slightly modified the expression to: “In clinical practice, when measuring LA volume, maximal LV volume is typically assessed.” (line 129-130, green highlight) Additionally, we described one research where it was reported that in HCM patients, minimal LA volume exhibited superior predictive capabilities for clinical outcomes (heart failure hospitalization, stroke, and all-cause mortality) compared to maximal LA volume. (line 133-136, green highlight)

Line 138-140 A recent meta-analysis reported that LAV and LAVI were higher in patients with AF recurrence following AF ablation compared to patients without AF recurrence. (19)

Abbreviation LAV (LA volume, I guess) has not been explained before.  

=>Response: Thank you for your critical comment. "I have removed the term 'LAV' as it was only used in this sentence, and I have revised it to 'LA volume' throughout the text. (line 141-142, green highlight)